# The effects of mindfulness-based interventions on symptoms of depression, anxiety, and cancer-related fatigue in oncology patients: A systematic review and meta-analysis

**Ellentika Chayadi, Naomi Baes[ID], Litza Kiropoulos[ID]***

Faculty of Medicine, Dentistry, and Health Sciences, Melbourne School of Psychological Sciences, The University of Melbourne, Melbourne, Australia

* litzak@unimelb.edu.au

## Abstract

### Objective

Mindfulness-based interventions (MBIs) are increasingly being integrated into oncological treatment to mitigate psychological distress and promote emotional and physical well-being. This review aims to provide the most recent evaluation of Mindfulness-Based Stress Reduction (MBSR), Mindfulness-Based Cognitive Therapy (MBCT), and Mindfulness-Based Cancer Recovery (MBCR) treatments, in reducing symptoms of depression, anxiety and CRF in oncology populations.

### Methods

A search using the following search terms was conducted: (mindful* OR mindfulness* OR mindfulness-based* OR MBI* OR MBCT OR MBSR OR MBCR) AND (Oncol* OR cancer OR neoplasm OR lymphoma OR carcinoma OR sarcoma) to obtain relevant publications from five databases: PsycINFO, PubMed, Embase, and MEDLINE by EC, and ProQuest Dissertations & Theses Global from January 2000 to February 2022. 36 independent studies ($n = 1677$) were evaluated for their overall effect sizes (using random-effects models), subgroup analyses, and quality appraisals. Evaluations were performed separately for non-randomized ($K = 20$, $n = 784$) and randomized controlled trials ($K = 16$, $n = 893$).

### Results

The results showed that MBIs have significant medium effects in reducing symptoms of depression (Hedges' $g = 0.43$), anxiety (Hedges' $g = 0.55$) and CRF (Hedges' $g = 0.43$), which were maintained at least three months post-intervention. MBIs were also superior in reducing symptoms of anxiety (Hedges' $g = 0.56$), depression (Hedges' $g = 0.43$), and CRF (Hedges' $g = 0.42$) in oncology samples relative to control groups. The superiority of MBIs to control groups was also maintained at least three months post-intervention for anxiety and

**Data Availability Statement:** All relevant data are within the paper and its Supporting Information files.

**Funding:** The author(s) received no specific funding for this work.

**Competing interests:** The authors have declared that no competing interests exist.

CRF symptoms, but not for depressive symptoms. The risk of bias of the included studies were low to moderate.

## Conclusions

This review found that MBIs reduced symptoms of depression, anxiety and CRF in oncology populations.

## Systematic review registration

PROSPERO: International Prospective Register of Systematic Reviews: CRD42020143286.

## 1. Introduction

Depression, anxiety and cancer-related fatigue (CRF) often accompany the diagnosis of cancer and its treatment. CRF is now recognized as one of the most common and distressing adverse effects of cancer and cancer treatment [1–3]. More than 70% of cancer patients experiencing CRF [4] discontinue treatment [4] and CRF persists in 25–30% of patients for 5 years and longer after successful treatment [5, 6]. Up to 38% of people diagnosed with cancer also present with clinically significant levels of depressive and anxiety symptoms in the first five years after diagnosis [7, 8] and these persist despite recovery from cancer [9–11]. Depression and anxiety can interfere with treatment adherence and the ability to cope with cancer, which in turn has been shown to exacerbate cancer progression [12]. Some research also suggests that depressive and anxiety symptoms are associated with higher cancer mortality [13–15]. Considering the possible bidirectional relationship between cancer and depressive and anxiety symptoms, it is important to integrate psychological support to cancer treatment [8, 16, 17].

The use of mindfulness-based interventions (MBIs) to mitigate depression and anxiety and promote emotional and physical well-being in cancer patients has become increasingly popular [8, 10, 16]. MBIs include Mindfulness-Based Stress Reduction [18] (MBSR), Mindfulness-Based Cognitive Therapy [19] (MBCT) and Mindfulness-Based Cancer Recovery [20] (MBCR). The MBSR program guides individuals to focus on their bodily sensations and acknowledge any discomfort without interpretation, elaboration, or evaluation as they engage in different mindfulness practices [18, 21]. The MBSR framework was further refined to include aspects of cognitive therapy, including recognizing and disengaging from reactive, analytic, and problem-solving thoughts, giving rise to MBCT [8, 10, 17, 19]. The cancer-specific adaptation of MBSR is known as MBCR [20]. Together, MBIs allow individuals to recognize, accept, and disengage from unpleasant physical sensations and dysfunctional thought process by directing attention to experience as it unfolds [8, 10, 18, 19]. In doing so, individuals also develop skills to counter reactive avoidance behavior and ruminative thought process that are akin to the development and relapse of anxiety and depression [8, 10, 17, 22].

In the past ten years, there have been three systematic reviews [17, 23, 24] and seven meta-analyses [9, 25–30] that have explored the effectiveness of MBIs in patients with cancer, but none have examined the effects of MBCR (see S1 Table). While most of the reviews looked into depression and anxiety as primary outcomes, only one meta-analysis explored the effects of MBIs on CRF [28] and this was in a breast cancer population. Furthermore, some of these reviews were conducted using a narrative method with minimal attempt to quantify the results

from the included studies [17]. While Ledesma and Kumano [25] explored the effectiveness of MBSR in improving physical well-being in a general cancer population and attempted to quantify the results of the included studies, this review was conducted in 2009 and included studies prior to 2007. Similarly, two of the latest reviews [8, 27] examining and quantifying the effectiveness of MBIs in reducing symptoms of depression and anxiety in cancer patients were undertaken in 2012 and 2015. Since then, more studies examining the effects of MBIs in reducing symptoms of depression, anxiety and CRF in patients with cancer have been published. More recently, Haller et al. [28] undertook a review of studies to examine the effectiveness of MBSR and MBCT on CRF in breast cancer populations and found these MBIs were effective in reducing CRF. Similarly, Rush et al. [23] found MBSR reduced stress symptoms in patients with cancer, but this review did not include studies using MBCT and MBCR.

It is important to consider the benefits of MBI for depression, anxiety and CRF in patients with cancer and the patient and treatment characteristics associated with more effective MBIs. The current review is the first to determine the ES of MBIs to treat depression, anxiety and CRF in patients with cancer. We considered studies examining the effectiveness of MBIs focusing on adult cancer patients published after 2007 that focused on the treatment of depression, anxiety and CRF. We hypothesized that: 1) MBIs are effective in reducing depressive and anxiety symptoms and CRF at post-intervention; 2) MBIs are effective in reducing depressive and anxiety symptoms and CRF and these reductions will be sustained at a 3-month follow-up period; 3) MBIs, compared to control conditions, are more effective in reducing depressive and anxiety symptoms and CRF at post-intervention; and 4) reductions in depressive and anxiety symptoms and CRF will be sustained at a 3-month follow-up period.

## 2. Method

### 2.1 Search strategy

The study adhered closely to the Preferred Reporting Items for Systematic Review and Meta-Analysis (PRISMA [31, 32]) guidelines and was registered with the International Prospective Register of Systematic Reviews (PROSPERO; CRD42020143286). Initially, a limited search of PsycINFO was conducted to identify relevant keywords contained in titles, abstract, and subject descriptors of papers. The resultant keywords found in this limited search are: '*mindfulness*', '*MBSR*', '*MBCT*', '*MBCR*', '*cancer*', '*oncology*', '*depression*', '*depressive symptoms*', '*anxiety*', and '*cancer-related fatigue*'. Search terms were developed in consultation with a medical librarian with the following search terms used: (mindful* OR mindfulness* OR mindfulness-based* OR MBI* OR MBCT OR MBSR OR MBCR) AND (Oncol* OR cancer OR neoplasm OR lymphoma OR carcinoma OR sarcoma) to obtain relevant publications from five databases: PsycINFO, PubMed, Embase, and MEDLINE by EC, and ProQuest Dissertations & Theses Global by NB. Searches were conducted from January 2000 to February 2022. In addition, EC manually screened all reference sections in articles retrieved for relevant manuscripts.

### 2.2 Study inclusion and exclusion criteria

**2.2.1 Type of studies.** Studies were included if they reported on the effects of MBSR, MBCT, and/or MBCR for patients with cancer and were reported in the English language. Studies included cross-sectional and longitudinal studies, pre- and post- treatment trials, randomized and non-randomized controlled trials, one thesis [24] and one doctoral dissertation [31]. Studies that did not report baseline data, case reports, reviews and papers not written in English were excluded.

**2.2.2 Participants.** Participants included individuals aged 18 years or above with a current or past diagnosis of cancer. Studies that included both patients with cancer and caregivers were included only if they provided a separate data set for patients with cancer, otherwise they were excluded.

**2.2.3 Interventions.** Interventions included were MBSR [18], MBCT [19], and/or MBCR [20]. Tailored MBIs for patients with cancer were also included if they did not vary significantly from the original MBIs.

**2.2.4 Outcomes.** Outcomes included screening and diagnostic measures of depression, anxiety and CRF and the mean and standard deviations of depression, anxiety and CRF.

## 2.3 Study selection

**2.3.1 Title and abstract screening.** Removal of duplicates from the resultant search from four data bases was undertaken by EC. NB searched the ProQuest Dissertations & Theses Global database. Titles, abstracts, and subject descriptors of all the relevant search results were independently screened for relevancy by five researchers, EC, NR, and GW, and LK and NB (for theses and dissertations) and were included for full-text review. Discrepancies were resolved via discussion and consensus.

**2.3.2 Full-text article review.** Full-text articles of relevant studies were independently reviewed against the eligibility criteria by five researchers (EC, GW, LK, NB, and NR). EC reviewed all relevant studies while NC and GW each reviewed half of the relevant studies (excluding theses and dissertations). LK and NB reviewed all eligible full text articles from the ProQuest Dissertations & Theses Global database. Discrepancies were resolved via discussion and consensus. Articles with missing information (e.g., means and standard deviations for anxiety, depression, or CRF measures) were identified and authors were contacted to request these statistics.

## 2.4 Data extraction

The following information from the included studies were extracted by EC and NB (for theses and dissertations): 1) study characteristics (including the design of the study, comparison conditions (if any), and the number of participants in each group); 2) intervention characteristics (including type of mindfulness-based intervention carried out, comparator group, the number of intervention sessions, the duration of instructional period); 3) sample characteristics (including age, gender, specific cancer diagnosis and stages); 4) outcome variables (including the questionnaire used to measure depressive and anxiety symptoms and CRF, the mean and standard deviations for these symptoms at pre-, post- and at least 3-month post-intervention).

## 2.5 Appraisal of methodical quality

Researchers (EC, NB, NR and GW) appraised the methodical quality independently using two different quality assessment scales, the Cochrane Risk of Bias V2 (RoB 2) [33] tool for randomized trials and the Risk of Bias in Non-randomized Studies of Interventions (ROBINS-I) [34] tool for non-randomized studies. The RoB 2 assesses biases in RCTs in five domains that are believed to affect the quality of the results including: 1) the randomization process; 2) deviations from intended interventions; 3) missing outcome data; 4) measurement of the outcome; and 5) selection of the reported result [33]. The ROBINS-I was used to assess the methodical quality of non-randomized trials [35]. Its assessment is divided into seven domains, addressing issues at baseline, during and post-interventions in the following domains: 1) confounders, 2) selection of participants into the study, 3) classification of interventions, 4) deviations from intended interventions, 5) missing data, 6) measurement of outcomes, and 7) selection of the

reported result [34]. Three researchers examined inter-rater reliability for both scales using the Kappa statistic [36] for all studies except the doctoral dissertation [31], which NB appraised using the RoB 2 tool for randomized trials.

## 2.6 Statistical methods

Standardized Mean Differences (SMDs) based on Hedges' g formula for continuous measures of depression, anxiety and CRF were used as Effect Sizes (ESs) using random effect modelling [39]. Quantitative data syntheses were carried out separately for within-group and between-group differences. ES for long term effects were calculated using the change in depression, anxiety and CRF scores from baseline measurement to the last available follow-up period. Positive ES indicates the effectiveness of the MBIs in reducing symptoms of depression, anxiety and CRF. Meanwhile, negative ES indicates a higher depression, anxiety and CRF post-intervention scores relative to the baseline. The overall ESs were pooled across studies using the random-effects model employing Comprehensive Meta-Analysis software [37]. Subgroup meta-analyses were conducted for the different MBIs. Fail-safe $N$ statistics and calculating Eggers' test were also computed to detect publication bias in the included studies. Heterogeneity was examined using the $Q$ and $I^2$ statistics. For primary outcomes with statistically significant heterogeneity ($p < .05$), a meta-regression was conducted to investigate the effect of total intervention hours, percentage female, and participant's mean age as potential moderators of the observed effect.

## 3. Results

### 3.1 Study selection

Overall, 2011 studies were retrieved from electronic sources, 712 duplicates were removed, and 1175 studies were excluded at the title and abstract screening phase. 124 full-text articles were then reviewed and 88 were excluded leaving a total of 36 independent studies which were included for final review. Fig 1 displays the PRISMA flow chart for the systematic review and meta-analysis.

### 3.2 Description of included studies

Table 1 displays the summarized characteristics of the 36 articles that met inclusion criteria. Studies investigated the effect of MBSR ($K = 29$), MBCT ($K = 5$), MBCR ($K = 2$) on symptoms of depression, anxiety or CRF in cancer patients or survivors. A total of 1650 participants underwent MBSR ($n = 1362$), MBCT ($n = 151$), and MBCR ($n = 137$). Participants were patients with mixed cancers ($K = 17$), breast cancer ($K = 16$) lung cancer ($K = 1$), prostate cancer ($K = 1$), and thyroid cancer ($K = 1$). Sixteen studies provided follow-up data for at least three months post-intervention (range 3–12 months). The majority of participants were females (82.66%) with a mean age of 54.50 years. The average total number and duration of weekly MBI sessions were 7.36 sessions and 135.00 minutes respectively. Most studies included a half-day retreat ($K = 14$) and gentle yoga sessions ($K = 28$) as part of the intervention. The average total length of the MBI across the studies was 20.49 hours. According to the studies that reported treatment adherence ($K = 24$), 89.25% of the participants completed at least 75% of all the MBI sessions (see S2 Table).

A total of 16 randomized controlled trials (RCTs) compared the effects of the MBSR ($K = 13$) or MBCT ($K = 3$) on symptoms of depression, anxiety and CRF to a control condition ($K = 15$) or psychoeducation ($K = 1$). The RCT sample ($n = 866$) consisted primarily of females (91.00%) and had a mean age of 53.76 years. There were no significant baseline differences in

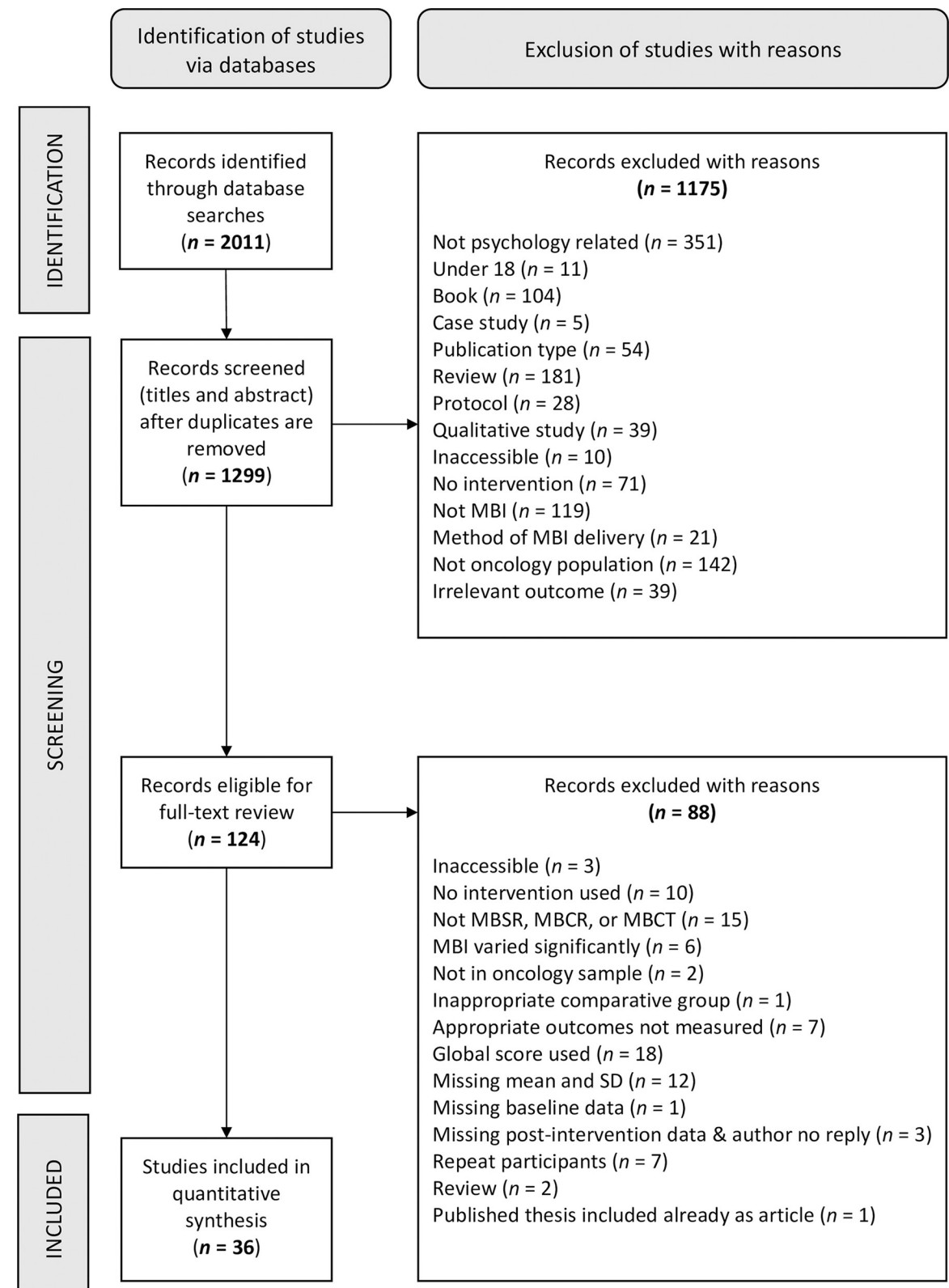

**Fig 1. PRISMA flow chart.**

**Table 1. Summarized characteristics of the 36 included studies.**

|  | Overall ($K$ = 36) | Non-RCTs ($K$ = 20) | RCTs ($K$ = 16) |
|---|---|---|---|
| Total sample in MBI groups | 1650 | 784 | 866 |
| Mean age | 54.50 | 54.99 | 53.76 |
| % Female | 82.66 | 80.99 | 91.00 |
| *No. of studies exploring* |  |  |  |
| MBSR | 29 | 16 | 13 |
| MBCT | 5 | 2 | 3 |
| MBCR | 2 | 2 | 0 |
| Average no. of sessions | 7.36 | 7.85 | 7.25 |
| Average duration for each session (minutes) | 135.00 | 124.50 | 118.13 |
| Average total length of intervention (hours) | 20.49 | 19.30 | 15.72 |

$K$ = number of studies that were included in the review; MBSR = Mindfulness-Based Stress Reduction;
MBCT = Mindfulness-Based Cognitive Therapy; MBCR = Mindfulness-Based Cancer Recovery; RCT = Randomised
Controlled Trial; nRCT = Non-Randomised Controlled Trial.

age, employment, education, cancer type, cancer stage, and time since diagnosis, depression, anxiety and CRF measures between participants in the MBI and comparison groups across the 16 studies. The average number of sessions was 7.25 and duration of the weekly sessions was 118.13 minutes. Some studies included a half-day retreat ($K$ = 4) and gentle yoga sessions ($K$ = 11) as part of the interventions. The average total hours of MBIs across the included studies was 15.72 hours.

## 3.3 Quality appraisal

**3.3.1 Randomized controlled trials.** Overall, the RCTs ($K$ = 16) showed low ($K$ = 7) to moderate ($K$ = 8) concerns for bias, with only one study indicating high risk of bias. The inter-rater reliability was examined using the Kappa statistic. The inter-rater reliability between the first author and the reviewers for the overall risk of bias in RCTs were substantial ($\kappa$ = 0.66, $p$ < .05). When assessing each of the specific domains of the RoB 2.0, the *Randomization Process* domain showed low ($K$ = 10) to some concerns ($K$ = 6). The inter-rater reliability for this domain was almost in perfect agreement ($\kappa$ = 0.86, $p$ < .05). The risk of bias from the *Deviations from Intended Interventions* domain was relatively low ($K$ = 13) with only three studies showing some concerns. The inter-rater reliability for this domain was substantial ($\kappa$ = 0.72, $p$ < .05). Similarly, the *Missing Outcome Data* domain showed low concerns ($K$ = 13) for most studies, two studies indicating some concerns, and one study indicating high concern. The inter-rater reliability for this domain was substantial ($\kappa$ = 0.66, $p$ < .05). The *Measurement of The Outcome* domain showed a relatively low ($K$ = 12) with only three studies showing some concerns. The inter-rater reliability for this domain was moderate ($\kappa$ = 0.42, $p$ < .05). The *Selection of The Reported Result* domain was of low concern ($K$ = 12) with only four studies showing some concerns. The inter-rater reliability for this domain was fair ($\kappa$ = 0.29, $p$ < .05). Fig 2 presents the risk of bias for all 16 RCTs.

**3.3.2 Non-randomized controlled trials.** The non-RCTs showed an overall moderate ($K$ = 12) risk of bias with 6 studies indicating low risk of bias and 2 studies showing serious risk of bias. The inter-rater reliability between the first author and the second reviewers for the overall risk of bias in non-RCTs were moderate ($\kappa$ = 0.46, $p$ < .05). The risk of bias for the specific domains, such as *Confounding* domain were widely spread between low ($K$ = 10), moderate ($K$ = 4) and serious ($K$ = 16) with fair agreement ($\kappa$ = 0.31, $p$ < .05). The *Selection of*

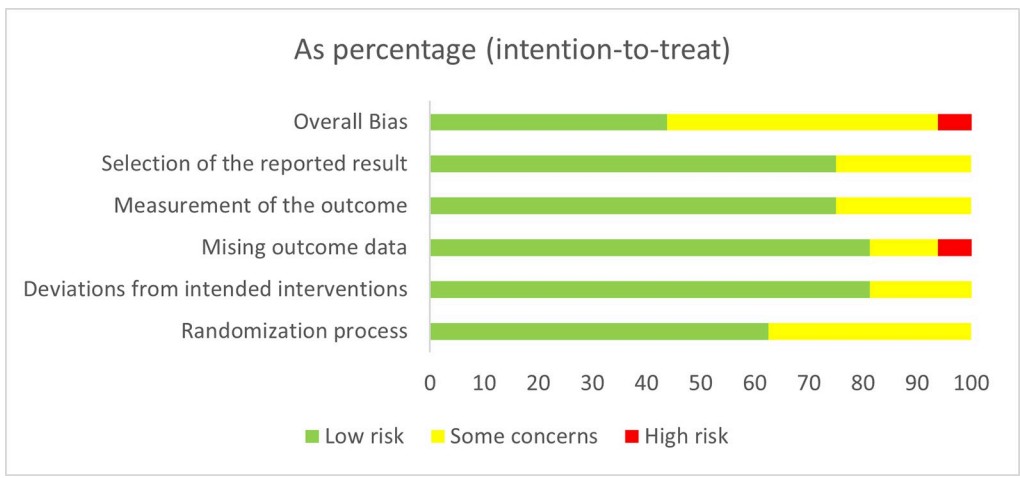

**Fig 2. RoB 2 assessment of risk of bias in RCTs.**

*Participants* domain showed mainly low ($K = 15$) risk of bias with only five showing moderate concerns for bias. The inter-rater reliability for this domain was moderate ($\kappa = 0.42$, $p < .05$). The *Classification of Interventions* domain showed low risk of bias ($K = 20$) with a substantial inter-rater reliability ($\kappa = 0.78$, $p < .05$). The *Deviations from Intended Interventions* domain showed mainly low ($K = 18$) risk of bias with only two studies indicating a moderate risk, with a moderate agreement ($\kappa = 0.55$, $p < .05$). The *Missing Data* domain showed a widespread of risk between low ($K = 8$), moderate ($K = 8$), and serious ($K = 4$) risk of bias. The inter-rater reliability for this domain was fair ($\kappa = 0.37$, $p < .05$). The *Measurement of Outcomes* domain showed mainly low risk of bias ($K = 19$) with only one study showing moderate risk of bias. The inter-rater reliability for this domain was substantial ($\kappa = 0.66$, $p < .05$). Finally, the *Selection of the Reported Result* showed mainly low risk of bias ($K = 16$) with only three studies indicating moderate risk of bias and one study showing serious risk of bias. The inter-rater reliability for this domain was moderate ($\kappa = 0.47$, $p < .05$). Fig 3 shows the risk of bias for non-RCT studies.

## 3.4 Data synthesis for within-group differences (pre-post effects)

**3.4.1 Overall ESs.** All 36 studies were included to determine the impact of the MBIs on symptoms of depression, anxiety and CRF pre- and post- intervention (see S3 Table). The

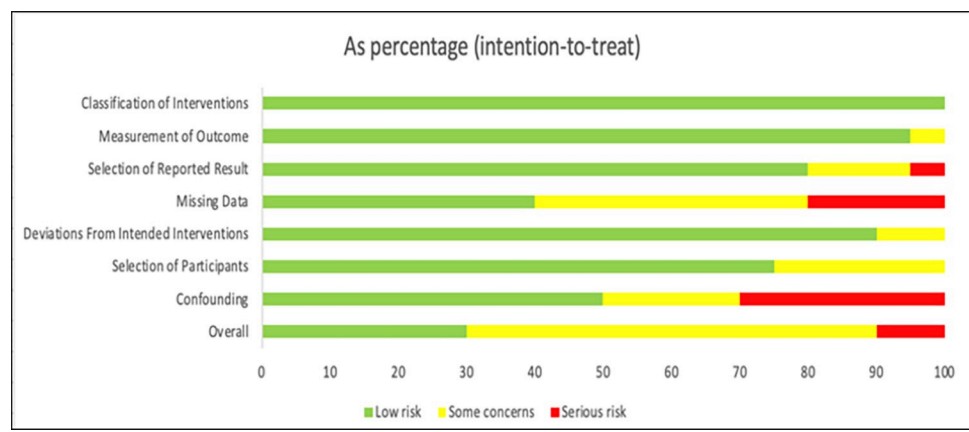

**Fig 3. ROBINS-I assessment of risk of bias in non-RCTs.**

**Table 2. Summary of the overall within group effect sizes.**

| Outcome Measures | Sample Size | | | Effect Size Estimate | | | Heterogeneity | | |
|---|---|---|---|---|---|---|---|---|---|
| | $K$ | $n$ | | Hedges' $g$ | 95% CI | $p$ | $Q$ | $p$ | $I^2$ (%) |
| Pre- and Post- differences | | | | | | | | | |
| Anxiety | 28 | 1306 | | 0.55 | [0.42, 0.68] | < 0.01 | 80.48 | < 0.01 | 66.45 |
| Depression | 31 | 1483 | | 0.43 | [0.31, 0.54] | < 0.01 | 93.43 | < 0.01 | 67.89 |
| CRF | 23 | 1244 | | 0.43 | [0.28, 0.59] | < 0.01 | 138.52 | < 0.01 | 84.84 |
| Follow-Up | | | | | | | | | |
| Anxiety | 14 | 759 | | 0.73 | [0.59, 0.87] | < 0.01 | 31.77 | < 0.01 | 59.08 |
| Depression | 14 | 847 | | 0.49 | [0.36, 0.63] | < 0.01 | 42.35 | < 0.01 | 69.30 |
| CRF | 12 | 814 | | 0.46 | [0.33, 0.58] | < 0.01 | 35.01 | < 0.01 | 68.63 |

*CRF* = cancer-related fatigue; *K* = number of studies that were included in the review; *n* = number of participants; CI = confidence interval.

overall pooled ESs for the pre- and post- intervention scores in both RCTs and non-RCTs showed a significant reduction in symptoms of depression ($K$ = 31, Hedges' $g$ = 0.43, S. E = 0.037, 95% CI: [0.36, 0.51], $p$ < 0.01), anxiety ($K$ = 28, Hedges' $g$ = 0.55, S.E = 0.067, 95% CI: [0.42, 0.68], $p$ < 0.01) and CRF ($K$ = 23, Hedges' $g$ = 0.46, S.E = 0.041, 95% CI: [0.22, 0.38], $p$ < 0.01). Forest plots display the within-group post-intervention effect sizes for anxiety (S1 Fig), depression (S3 Fig) and CRF (S5 Fig).

**3.4.2 Follow-up ESs.** Studies provided data for symptoms of depression ($K$ = 15), anxiety ($K$ = 14) and CRF ($K$ = 12) scores at least three months post-intervention (See S4 Table). The overall pooled ESs showed a significant reduction in symptoms of depression (Hedges' $g$ = 0.49, S.E = 0.10, 95% CI: [0.38, 0.77], $p$ < 0.01), anxiety (Hedges' $g$ = 0.73, S.E = 0.072, 95% CI: [0.59, 0.87], $p$ < 0.01) and CRF (Hedges' $g$ = 0.46, S.E = 0.066, 95% CI: [0.33, 0.59], $p$ < 0.01) at least three months post-intervention. Table 2 shows the summary of the overall within-group effect sizes for post-intervention and follow-up studies. Forest plots are shown for the within-group effect sizes of MBIs at follow-up intervention timepoints for anxiety (S2 Fig), depressive (S4 Fig), and CRF (S6 Fig) symptoms.

**3.4.3 Heterogeneity.** As seen in Table 2, there was a significant moderate to high between-study heterogeneity for the pre- and post- intervention scores for depression ($Q$ = 93.043, $p$ < 0.01, $I^2$ = 67.89), anxiety ($Q$ = 80.48, $p$ < 0.01, $I^2$ = 66.45) and CRF ($Q$ = 138.52, $p$ < 0.01, $I^2$ = 84.84) in all 36 studies. A similar pattern was also found for the follow-up effect of MBIs on symptoms of depression ($Q$ = 42.35, $p$ < 0.01, $I^2$ = 69.30), anxiety ($Q$ = 31.77, $p$ < 0.01, $I^2$ = 59.08) and CRF ($Q$ = 35.01, $p$ < 0.01, $I^2$ = 68.63).

**3.4.4 Publication bias.** There was no evidence of publication bias for the studies comparing the change in depression (Fail-safe $N$ = 925, Egger's $t(29)$ = 0.02, $p$ = 0.985), anxiety (Fail-safe $N$ = 1259, Egger's $t(26)$ = 0.51, $p$ = 0.611) and CRF (Fail-safe $N$ = 215, Egger's $t(20)$ = 0.59, $p$ = 0.563) scores before and after the MBIs. Similarly, no evidence of publication bias was found for the studies comparing the long-term (at least 3 months post MBI) change in depression (Fail-safe $N$ = 361, Egger's $t(12)$ = 1.06, $p$ = 0.308) and CRF (Fail-safe $N$ = 184, Egger's $t(10)$ = 0.21, $p$ = 0.840) scores. However, evidence of publication bias in studies examining the long-term effects of MBI on symptoms of anxiety (Fail-safe $N$ = 621, Egger's $t(12)$ = 2.22, $p$ = 0.046) were found.

**3.4.5 Subgroup analyses.** As shown in Table 3, the subgroup analyses revealed that MBSR, MBCT and MBCR were all effective in reducing symptoms of anxiety (MBSR–Hedges' $g$ = 0.54, 95% CI: [0.39, 0.69], $p$ < 0.01; MBCT–Hedges' $g$ = 0.81, 95% CI: [0.27, 1.35], $p$ < 0.01; MBCR–Hedges' $g$ = 0.51, 95% CI: [0.22, 0.81], $p$ < 0.01). However, for CRF, only

**Table 3. Summary of the overall subgroup analyses for within group effect sizes.**

| Outcome Measure | MBI | K | Effect Size Estimate | | | Heterogeneity | | |
|---|---|---|---|---|---|---|---|---|
| | | | Hedges' $g$ | 95% CI | $p$ | Q | $p$ | $I^2$ (%) |
| Pre- and Post- differences | | | | | | | | |
| Anxiety | MBSR | 23 | 0.54 | [0.39, 0.69] | < 0.01 | 63.86 | < 0.01 | 65.55 |
| | MBCT | 4 | 0.81 | [0.27, 1.35] | < 0.01 | 8.15 | < 0.05 | 63.19 |
| | MBCR | 1 | 0.51 | [0.22, 0.81] | < 0.01 | 0.0 | 1.00 | 0.00 |
| Depression | MBSR | 25 | 0.41 | [0.28, 0.55] | < 0.01 | 60.85 | < 0.01 | 60.56 |
| | MBCT | 4 | 0.43 | [-0.53, 1.40] | 0.378 | 26.24 | < 0.01 | 88.57 |
| | MBCR | 2 | 0.47 | [0.23, 0.71] | < 0.01 | 0.24 | 0.624 | 0.00 |
| CRF | MBSR | 20 | 0.21 | [-0.01, 0.43] | 0.066 | 112.35 | < 0.01 | 83.09 |
| | MBCT | 1 | 1.30 | [0.90, 1.70] | < 0.01 | 0.00 | 1.00 | 0.00 |
| | MBCR | 1 | 0.37 | [0.08, 0.66] | 0.012 | 0.00 | 1.00 | 0.00 |
| Follow-Up | | | | | | | | |
| Anxiety | MBSR | 10 | 0.67 | [0.52, 0.82] | < 0.01 | 13.79 | 0.130 | 34.70 |
| | MBCT | 4 | 1.18 | [0.77, 1.59] | < 0.01 | 4.69 | 0.196 | 36.00 |
| Depression | MBSR | 11 | 0.48 | [0.34, 0.62] | < 0.01 | 15.29 | 0.122 | 34.59 |
| | MBCT | 3 | 0.87 | [-0.02, 1.75] | 0.056 | 12.13 | < 0.01 | 83.51 |
| CRF | MBSR | 11 | 0.36 | [0.22, 0.50] | < 0.01 | 15.04 | 0.131 | 33.52 |
| | MBCT | 1 | 1.30 | [0.91, 1.69] | < 0.01 | 0.00 | 1.00 | 0.00 |

CRF = cancer-related fatigue; K = number of studies that were included in the review; n = number of participants; CI = confidence interval; MBSR = Mindfulness-Based Stress Reduction; MBCT = Mindfulness-Based Cognitive Therapy; MBCR = Mindfulness-Based Cancer Recovery.

MBCT (Hedges' $g$ = 1.30, 95% CI: [0.90, 1.70], $p$ < 0.01) and MBCR (Hedges' $g$ = 0.37, 95% CI: [0.08, 0.66], $p$ < 0.012) significantly reduced symptoms of CRF. For depression, only MBSR (Hedges' $g$ = 0.41, 95% CI: [0.28, 0.55], $p$ < 0.01) and MBCR (Hedges' $g$ = 0.47, 95% CI: [0.23, 0.71], $p$ < 0.01) significantly reduced symptoms of depression, not MBCT.

Longitudinally, the subgroup analyses revealed that MBSR and MBCT were effective in reducing symptoms of depression (MBSR–Hedges' $g$ = 0.48, 95% CI: [0.34, 0.62], $p$ < 0.01; MBCT–Hedges' $g$ = 0.87, 95% CI: [-0.02, 1.75], $p$ = 0.56), anxiety (MBSR–Hedges' $g$ = 0.67, 95% CI: [0.52, 0.82], $p$ < 0.01; MBCT–Hedges' $g$ = 1.18, 95% CI: [0.77, 1.59], $p$ < 0.01) and CRF (MBSR–Hedges' $g$ = 0.36, 95% CI: [0.22, 0.50], $p$ < 0.01; MBCT–Hedges' $g$ = 1.30, 95% CI: [0.91, 1.69], $p$ < 0.01). However, it is important to note the relatively fewer studies that examined the effects of MBCT and MBCR on depression, anxiety and CRF compared to MBSR. No studies examined the long-term effects of MBCR on symptoms of depression, anxiety or CRF.

Similar to the overall effects, the subgroup analyses also revealed mostly significant heterogeneity between studies. Exceptions were seen for depression, anxiety and CRF scores in studies using MBCR as an intervention. However, the number of studies using MBCR as an intervention was significantly lower (K = 1–2), limiting the generalizability of the result. Heterogeneity between studies was low to moderate for the follow-up effect of MBSR intervention on its effects on symptoms of depression (Q = 15.29, $p$ = 0.122, $I^2$ = 34.59), anxiety (Q = 13.79, $p$ = 0.130, $I^2$ = 34.70) and MBCT interventions on symptoms of anxiety (Q = 4.69, $p$ = 0.196, $I^2$ = 36.00), suggesting that these studies are homogeneous.

**3.4.6 Meta-regression.** Meta-regressions were undertaken to account for the significant heterogeneity in the included studies. Moderators comprised of the total intervention hours, percentage of female participants, and participant's mean age. These moderators were not significant predictors of levels of depression, anxiety and CRF.

**Table 4. Summary of the overall between group effect sizes in RCTs.**

| Outcome Measures | Sample Size | | Effect Size Estimate | | | Heterogeneity | | |
|---|---|---|---|---|---|---|---|---|
| | $K$ | $n$ | Hedges' $g$ | 95% CI | $p$ | $Q$ | $p$ | $I^2$ (%) |
| Between Group Differences | | | | | | | | |
| Anxiety | 12 | 1269 | 0.56 | [0.35, 0.77] | < 0.01 | 31.80 | < 0.01 | 65.41 |
| Depression | 12 | 1290 | 0.43 | [0.13, 0.72] | < 0.01 | 56.42 | < 0.01 | 80.50 |
| CRF | 9 | 1167 | 0.42 | [0.18, 0.66] | < 0.01 | 38.19 | < 0.01 | 79.05 |
| Follow- Up | | | | | | | | |
| Anxiety | 5 | 745 | 0.58 | [0.21, 0.95] | < 0.01 | 22.20 | < 0.01 | 81.99 |
| Depression | 5 | 911 | 0.23 | [0.15, 0.43] | 0.088 | 37.90 | < 0.01 | 89.45 |
| CRF | 5 | 807 | 0.33 | [0.18, 0.47] | < 0.01 | 4.28 | 0.369 | 6.56 |

*CRF* = cancer related fatigue; *K* = number of studies that were included in the review; *n* = number of participants; CI = confidence interval.

### 3.5 Data synthesis for between-group differences

**3.5.1 Overall ESs.** A total of 16 RCTs compared the effects of the MBSR ($K$ = 13) or MBCT ($K$ = 3) on symptoms of depression, anxiety and CRF to a control condition ($K$ = 15) or psychoeducation ($K$ = 1) [38]. The overall pooled ESs comparing MBIs to control or psychoeducation revealed a significant superiority of MBIs in reducing symptoms of depression ($K$ = 12, Hedges' $g$ = 0.43, S.E = 0.059, 95% CI: [0.32, 0.55], $p$ < 0.01), anxiety ($K$ = 12, Hedges' $g$ = 0.56, S.E = 0.107, 95% CI: [0.35, 0.77], $p$ < 0.01) and CRF ($K$ = 10, Hedges' $g$ = 0.42, S.E = 0.060, 95% CI: [0.18, 0.66], $p$ < 0.01). Table in S5 Table displays the between group effect sizes for MBIs compared to control conditions.

**3.5.2 Follow-up ESs.** 15 studies provided data for symptoms of depression ($K$ = 5), anxiety ($K$ = 5) and CRF ($K$ = 5) scores at least three months post-intervention. The overall pooled effect sizes (ESs) showed that MBIs significantly reduced symptoms of anxiety (Hedges' $g$ = 0.58, S.E = 0.190, 95% CI: [0.21, 0.95], $p$ < 0.01) and CRF (Hedges' $g$ = 0.33, S.E = 0.074, 95% CI: [0.18, 0.47], $p$ < 0.01) at least three months after the baseline measurement relative to control group. No differences in MBIs and control group was found for depressive symptoms (Hedges' $g$ = 0.23, S.E = 0.230, 95% CI: [0.15, 0.43], $p$ = 0.369). Table 4 summarizes the overall within-group effect sizes comparing the effects of MBIs to control conditions. The table in S6 Table shows the between group effect sizes for MBIs compared to control conditions. The forest plots of the between-group effects of MBIs show depressive, anxiety and CRF symptoms at post intervention (S7, S9 and S11 Figs) and follow-up (S8, S10 and S12 Figs) intervention.

**3.5.3 Heterogeneity.** As shown in Table 4, there was a moderate to high between-study heterogeneity for the analyses of depression ($Q$ = 56.42, $p$ < 0.01, $I^2$ = 80.50), anxiety ($Q$ = 31.80, $p$ < 0.01, $I^2$ = 65.41) and CRF ($Q$ = 38.19, $p$ < 0.01, $I^2$ = 79.05). A similar pattern was also found for the follow-up effect of MBIs on symptoms of depression ($Q$ = 37.90, $p$ < 0.01, $I^2$ = 89.45) and anxiety ($Q$ = 22.2, $p$ < 0.01, $I^2$ = 81.99). However, the Q and $I^2$ statistics indicate homogeneity in studies analysing CRF ($Q$ = 4.28, $p$ = 0.369, $I^2$ = 6.56) relative to controls.

**3.5.4 Publication bias.** There was no evidence of publication bias for the RCTs comparing the change in anxiety (Fail-safe $N$ = 224, Egger's $t$(10) = 1.75, $p$ = 0.111) and depression (Fail-safe $N$ = 136, Egger's $t$(10) = 0.03, $p$ = 0.976), or CRF (Fail-safe $N$ = 57, Egger's $t$(7) = 0.09, $p$ = 0.928) scores in MBI groups to control conditions. Similarly, there was no evidence of publication bias for the RCTs comparing the long-term change (at least three months post MBI) in depression (Fail-safe $N$ = 24, Egger's $t$(3) = 0.85, $p$ = 0.459), anxiety (Fail-safe $N$ = 55, Egger's $t$(3) = 1.26, $p$ = 0.296) and CRF (Fail-safe $N$ = 23, Egger's $t$(3) = 0.61, $p$ = 0.588) scores in the MBI groups compared to the control groups.

**Table 5. Summary of the overall subgroup analyses for between group effect sizes.**

| Outcome Measure | MBI | $K$ | Effect Size Estimate | | | Heterogeneity | | |
|---|---|---|---|---|---|---|---|---|
| | | | Hedges' $g$ | 95% CI | $p$ | $Q$ | $p$ | $I^2$ (%) |
| Between Group Differences | | | | | | | | |
| Anxiety | MBSR | 10 | 0.54 | [0.31, 0.77] | < 0.01 | 29.61 | < 0.01 | 69.61 |
| | MBCT | 2 | 0.71 | [0.14, 1.28] | 0.014 | 1.54 | 0.220 | 35.07 |
| Depression | MBSR | 10 | 0.43 | [0.13, 0.74] | < 0.01 | 50.39 | < 0.01 | 82.12 |
| | MBCT | 2 | 0.92 | [0.43, 1.41] | < 0.01 | 4.57 | 0.033 | 78.12 |
| CRF | MBSR | 8 | 0.27 | [-0.01, 0.54] | < 0.01 | 32.34 | < 0.01 | 78.36 |
| | MBCT | 1 | 0.91 | [0.43, 1.40] | 0.058 | 0.00 | 1.00 | 0.00 |

CRF = cancer-related fatigue; $K$ = number of studies that were included in the review; $n$ = number of participants; CI = confidence interval; MBSR = Mindfulness-Based Stress Reduction; MBCT = Mindfulness-Based Cognitive Therapy.

**3.5.5 Subgroup analyses.** Subgroup analyses were not carried out for the longer-term effects of MBIs on symptoms of anxiety, depression, and CRF relative to control conditions as studies examining the longer-term effects only included MBSR as an intervention. None of the RCTs investigated the effects of MBCR in reducing symptoms of depression, anxiety and CRF. As shown in Table 5, the subgroup analyses revealed that MBSR and MBCT were significantly effective in reducing symptoms of depression (MBSR–Hedges' $g$ = 0.43, 95% CI: [0.13, 0.74], $p$ < 0.01; MBCT–Hedges' $g$ = 0.92, 95% CI: [0.43, 1.41], $p$ < 0.01) and anxiety (MBSR–Hedges' $g$ = 0.54, 95% CI: [0.31, 0.77], $p$ < 0.01; MBCT–Hedges' $g$ = 0.71, 95% CI: [0.14, 1.28], $p$ < 0.01). For CRF, only MBCT was reported to be significantly effective in reducing symptoms (Hedges' $g$ = 0.27, 95% CI: [0.16, 0.66], $p$ < 0.01) relative to control conditions. However, it is important to note that there were relatively fewer studies that examined the effects of MBCT on depression, anxiety and CRF compared to MBSR.

Similar to the overall effects, the subgroup analyses also revealed mostly significant heterogeneity between studies, particularly for studies that implemented MBSR as an intervention. This suggests that there were significant differences between the studies that implemented MBSR for treatment of depression, anxiety and CRF. While MBCT showed a similar trend, exceptions were seen for studies using MBCT as an intervention for scores on anxiety ($Q$ = 1.54, $p$ = 0.220, $I^2$ = 35.07) and CRF. However, the heterogeneity results may not be meaningful, particularly for CRF, as the number of studies using MBCT as an intervention were substantially lower ($K$ = 1–2).

**3.5.6 Meta-regression.** The total intervention hours, female participants (%), and mean age were used as moderators. None of the moderators were significant predictors of levels of depression, anxiety and CRF in the RCTs.

## 4. Discussion

The current review provided the most current synthesis of 36 existing studies exploring the effects of MBIs, specifically MBSR, MBCT, and MBCR, in reducing symptoms of depression, anxiety and CRF in patients with cancer. While the analysis indicated heterogeneity in the included studies, risk of bias for the studies remained at low to moderate risk. The heterogeneity in the included studies may be explained due to the individual differences in the sample or in the delivery of MBIs. Samples varied due to different stages of cancer, cancer types, and cancer treatments. Furthermore, some studies only included women [22, 38–53] experiencing breast cancer. Other samples varied due to history of mental illness, use of anxiolytic or anti-depressant medications, familiarity with MBIs, trait mindfulness, and comorbid physical

conditions [8]. MBIs that were offered differed on factors such as total contact hours, qualifications and experience of the facilitators. However, while contact hours was not shown to be a significant moderator, these factors were not systematically and consistently reported in all studies, and therefore could not be adequately evaluated.

## 4.1 Effectiveness of MBIs

MBIs were shown to be effective in reducing symptoms of depression, anxiety and CRF in patients with cancer both at post- and follow-up intervention. The change in baseline to post-intervention showed significant medium overall effect size on anxiety (Hedges' $g$ = 0.55), which appeared to be relatively more effective at three-month follow-up (Hedges' $g$ = 0.72). The MBIs also had a small, but significant, effect on symptoms of depression (Hedges' $g$ = 0.43), which increased slightly at three-month follow-up (Hedges' $g$ = 0.49). Finally, MBIs were effective in reducing symptoms of CRF (Hedges' $g$ = 0.43) and the effects remained significant at three-month follow up (Hedges' $g$ = 0.46). Present results and size of effectiveness is consistent with the most recent meta-analyses that have used patients with cancer [8, 27]. To our knowledge, there is no previous meta-analysis comparing the effects of MBIs on CRF or the long-term effects of MBIs on depression, anxiety and CRF.

The subgroup analyses revealed that MBSR was effective in reducing symptoms of anxiety (Hedges' $g$ = 0.54) after the intervention and the effect was sustained at three months after the intervention (Hedges' $g$ = 0.67). MBSR was the most effective in reducing symptoms of anxiety. Similarly, MBSR was effective in reducing symptoms of depression at post-intervention (Hedges' $g$ = 0.41) and at three months after the intervention (Hedges' $g$ = 0.48). MBSR was also effective in reducing symptoms of CRF immediately after the intervention (Hedges' $g$ = 0.21) and its effect was relatively larger at three-month follow up (Hedges' $g$ = 0.36). In general, the effect of MBSR on symptoms of depression, anxiety and CRF was relatively larger three months after the intervention. This finding is consistent with meta-analyses that were conducted on the effects of MBSR in reducing symptoms of anxiety in oncology patients [8, 25, 28–30].

The analyses also showed a significantly large effect of MBCT in reducing symptoms of anxiety (Hedges' $g$ = 0.81) and CRF (Hedges' $g$ = 1.30) at post-intervention. The reductions in anxiety (Hedges' $g$ = 1.18) and CRF (Hedges' $g$ = 1.30) are also maintained at three-month follow-up. However, the data showed that MBCT was not effective in reducing symptoms of depression after the intervention. This is an unexpected finding as many studies have suggested the effectiveness of MBCT in reducing symptoms of depression [9, 12]. Nevertheless, findings may be impacted by the low number of studies testing the effectiveness of MBCT interventions. Considering that MBCT was derived from a model to reduce the risks of relapse and recurrence of major depressive episode [19], the longer term effectiveness of MBCT found in this study supports the original intention of the treatment. In comparison, MBCT appears to have the largest effectiveness in reducing symptoms of anxiety and CRF relative to MBSR and MBCR.

While MBCR appears to be effective in reducing symptoms of depression (Hedges' $g$ = 0.47), anxiety (Hedges' $g$ = 0.51) and CRF (Hedges' $g$ = 0.37), no study had explored its long-term effects. Although the analysis revealed a significant small to medium effect size of MBCR in reducing symptoms of depression, anxiety and CRF, it is important to note that there were only one to two studies that examined the effects of MBCR.

## 4.2 Effectiveness of MBIs relative to control groups

MBIs were found to be more effective in reducing symptoms of anxiety and CRF in oncology populations relative to control conditions at post-intervention and at least three months after

the intervention. While MBIs are more effective in reducing symptoms of depression relative to control conditions at post-intervention, its effectiveness was not maintained at three months post-intervention. Furthermore, while the effectiveness of MBIs in reducing symptoms of anxiety remains relatively stable at the three-month follow up period (Hedges' $g$ = 0.56 at post- and 0.58 at least three months post intervention), the effectiveness of MBIs in reducing symptoms of CRF became relatively smaller at three-month follow-up (Hedges' $g$ = 0.42 at post- to 0.33 at least three months post-intervention). This result is consistent with the finding of most recent meta-analyses that used patients with cancer and compared the effectiveness of MBIs in RCTs [12]. However, the current study showed a higher effect size of MBIs in reducing symptoms of depression and anxiety relative to the effect sizes (Hedges' $g$ = 0.37 and Hedges' $g$ = 0.44 respectively; small effect size) reported by Piet and colleagues [12].

Subgroup analyses were only done on MBSR and MBCT studies as there were no RCTs measuring the effects of MBCR on symptoms of depression, anxiety and CRF against a control condition. A subgroup analysis was not done for the longer-term effects of the MBIs as studies examining these only included MBSR as an intervention. The analyses suggested that MBSR was superior in reducing symptoms of depression (Hedges' $g$ = 0.43), anxiety (Hedges' $g$ = 0.54) and CRF (Hedges' $g$ = 0.27) relative to control conditions, with medium effect sizes found. Similarly, MBCT was also superior in reducing symptoms of depression (Hedges' $g$ = 0.92) relative to control conditions, but not in reducing symptoms of CRF. Nevertheless, it is important to note that there was only one study that examined the effect of MBCT on CRF scores, limiting the conclusions that can be made.

## 5 Conclusion

### 5.1 Study strengths

This review and meta-analysis has several strengths. It evaluated studies exploring the effects of MBIs on depressive and anxiety symptoms and CRF, including MBCR, at post- and follow-up intervention timepoints that have been conducted in the past ten years. It also included RCTs to establish the superiority of MBIs relative to control groups. The methodological quality of the RCTs and non-RCTs were also evaluated.

### 5.2 Limitations and future directions

There were a few limitations to this review. Firstly, while the review included participants with all types of cancer, there were variations in the cancer stage, time since diagnosis, treatment progress, and cancer prognosis between studies. Furthermore, studies included in this review did not specify if the patients with cancer had received MBIs in the past or received a psychological diagnosis or treatment. These factors may have influenced the recorded effectiveness of the MBIs in the studies and the generalisability of the findings to cancer patients with psychological disorders. Future research should consider exploring the effects of MBIs on patients with cancer with confirmed psychological diagnoses. It should also examine these factors as potential moderators of the effectiveness of MBIs. Secondly, little information is known about the qualifications of the facilitators in the studies which may affect the effectiveness of MBIs provided. Future studies should explore clinicians' experience, qualification, and engagement as a moderator of MBIs. Furthermore, considering that the meta-regression revealed that MBI contact hours was not a significant moderator future studies should explore the possibility of conducting MBIs with shorter duration or lower number of sessions. This may be beneficial for patients with cancer who do not have the time or the physical resources to participate in extensive MBI programs [54]. Thirdly, it is important to note that the number of studies exploring MBCT and MBCR are relatively smaller than studies exploring the effects of MBSR

and studies exploring the effects of MBIs on CRF. Thus, findings from this review for MBCT and MBCR and the effects of MBIs on CRF need to be interpreted with caution due to the small number of studies included. In addition, all search results were limited to studies written in the English language which may have influenced results. Future studies exploring the effects of MBCT and MBCR on depressive and anxiety symptoms and MBIs on CRF in cancer patients are required.

### 5.3 Clinical implications

From the present findings, it is evident that MBIs are effective in reducing symptoms of depression, anxiety and CRF in patients with cancer both at post- and follow-up timepoints relative to control groups. The findings also suggest that the use of MBIs in the care of patients with cancer is feasible. MBIs are also low-cost as the therapist to group ratio are typically small which is ideal for most health system environments which have constrained resources. Future studies should continue to examine the effectiveness and cost-effectiveness of providing MBIs to patients with cancer.

## Supporting information

**S1 Table. Summary of the reviews undertaken in the recent ten years.**
(DOCX)

**S2 Table. Summary of MBI in each study.**
(DOCX)

**S3 Table. Within group effect sizes for pre- and post-intervention scores.**
(DOCX)

**S4 Table. Longer term within group effect sizes.**
(DOCX)

**S5 Table. Between group effect sizes comparing MBIs to control condition.**
(DOCX)

**S6 Table. Longer term between group effect sizes of MBIs compared to control condition.**
(DOCX)

**S1 Fig. Within group differences–anxiety.**
(TIF)

**S2 Fig. Within group differences–anxiety follow-up.**
(TIF)

**S3 Fig. Within group differences–depression.**
(TIF)

**S4 Fig. Within group differences–depression follow-up.**
(TIF)

**S5 Fig. Within group differences–CRF.**
(PNG)

**S6 Fig. Within group differences–CRF follow-up.**
(TIF)

**S7 Fig. Between group differences–anxiety.**
(TIF)

**S8 Fig. Between group differences–anxiety follow-up.**
(TIF)

**S9 Fig. Between group differences–depression.**
(TIF)

**S10 Fig. Between group differences–depression follow-up.**
(TIF)

**S11 Fig. Between group differences–CRF.**
(TIF)

**S12 Fig. Between group differences–CRF follow-up.**
(TIF)

**S1 Appendix.**
(DOCX)

## Acknowledgments

We would like to thank Gabrielle Williams and Nicky Rickerby for reviewing the studies included in this review. We would also like to thank Patrick Condron for guidance with the search terms.

## Author Contributions

**Conceptualization:** Ellentika Chayadi, Litza Kiropoulos.

**Data curation:** Ellentika Chayadi, Naomi Baes, Litza Kiropoulos.

**Formal analysis:** Ellentika Chayadi, Naomi Baes, Litza Kiropoulos.

**Investigation:** Litza Kiropoulos.

**Methodology:** Naomi Baes, Litza Kiropoulos.

**Resources:** Litza Kiropoulos.

**Supervision:** Litza Kiropoulos.

**Writing – original draft:** Ellentika Chayadi, Litza Kiropoulos.

**Writing – review & editing:** Ellentika Chayadi, Naomi Baes, Litza Kiropoulos.

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
