## [Decision Letter · Decision Letter 0]

2 Dec 2021

PONE-D-21-18799The Effects of Mindfulness-Based Interventions on Symptoms of Depression, Anxiety, and Cancer-Related Fatigue in Oncology Patients: A Systematic Review and Meta-AnalysisPLOS ONE

Dear Dr. Kiropoulos,

Thank you for submitting your manuscript to PLOS ONE. After careful consideration, we feel that it has merit but does not fully meet PLOS ONE’s publication criteria as it currently stands. Therefore, we invite you to submit a revised version of the manuscript that addresses the points raised during the review process.

The systematic review summarizes the current finding on mindfulness stress reduction therapy in cancer related depression, anxiety and fatigue. The reviewer was pleased with the work but pointed out a few issues on the search strategy, excluded studies and biases, as well as the grey literature. Please address these comments (Please see the detailed in the Reviewer's comments). In addition to the comments from the reviewers, please find the editorial comments below:

1. In the Abstract: 

1.1 Search strategy, search strategy, key words and time line should be summarized in the Methods of the Abstract.

1.2 Please define your conclusion whether mindfulness reduces the symptoms from cancer or not.

2. In the Introduction:

Please clearly define these terms: Mindfulness-Based Stress Reduction18 (MBSR), Mindfulness-Based Cognitive Therapy (MBCT) and Mindfulness-Based Cancer Recovery (MBCR). 

3. in the Methods:

3.1 Define more clearly exclusion criteria. For example, I would imagine you exclude case reports, case series, reviews, etc. (and non-English literature?)

3.2 Did you do extensive search of the references for the reviews of full text articles? 

3.3 Please add information on Grey literature (see also Reviewer's comments).

4. In Results:

 Can we have a graph to show something like effect sizes from each modality?

5. In the Discussion:

5.1  Please add a discussion on the implications with other types of patients, end stage cancer v.s. cancer treatable/inremission

5.2 Please also discuss clinical trials or studies that may not be reported, grey literature, theses, etc.

We look forward to receiving your revised manuscript.

Kind regards,

Sompop Bencharit, DDS, MS, PhD, FACP

Academic Editor

PLOS ONE

Journal Requirements:

4. Please include your tables as part of your main manuscript and remove the individual files. Please note that supplementary tables (should remain/ be uploaded) as separate "supporting information" files.

Additional Editor Comments (if provided):

The systematic review summarizes the current finding on mindfulness stress reduction therapy in cancer related depression, anxiety and fatigue. The reviewer was pleased with the work but pointed out a few issues on the search strategy, excluded studies and biases, as well as the grey literature. Please address these comments (Please see the detailed in the Reviewer's comments). In addition to the comments from the reviewers, please find the editorial comments below:

1. In the Abstract:

1.1 Search strategy, search strategy, key words and time line should be summarized in the Methods of the Abstract.

1.2 Please define your conclusion whether mindfulness reduces the symptoms from cancer or not.

2. In the Introduction:

Please clearly define these terms: Mindfulness-Based Stress Reduction18 (MBSR), Mindfulness-Based Cognitive Therapy (MBCT) and Mindfulness-Based Cancer Recovery (MBCR).

3. in the Methods:

3.1 Define more clearly exclusion criteria. For example, I would imagine you exclude case reports, case series, reviews, etc. (and non-English literature?)

3.2 Did you do extensive search of the references for the reviews of full text articles?

3.3 Please add information on Grey literature (see also Reviewer's comments).

4. In Results:

Can we have a graph to show something like effect sizes from each modality?

5. In the Discussion:

5.1 Please add a discussion on the implications with other types of patients, end stage cancer v.s. cancer treatable/inremission

5.2 Please also discuss clinical trials or studies that may not be reported, grey literature, theses, etc.

Reviewers' comments:

Reviewer's Responses to Questions

**Comments to the Author**

1. Is the manuscript technically sound, and do the data support the conclusions?

Reviewer #1: Yes

2. Has the statistical analysis been performed appropriately and rigorously? 

Reviewer #1: I Don't Know

3. Have the authors made all data underlying the findings in their manuscript fully available?

Reviewer #1: Yes

4. Is the manuscript presented in an intelligible fashion and written in standard English?

Reviewer #1: Yes

5. Review Comments to the Author

Reviewer #1: Thank you for this well conducted systematic review and meta-analysis. I have just a few recommendations and questions:

Recommend more fully addressing item 7 of the PRISMA checklist to provide the full search strategy used for at least one database, including the keywords and controlled vocabulary with the Boolean operators used.

Regarding PRISMA checklist item 23c - discuss any limitations of the review processes used - Please address this item more fully. For example, note that search results were limited to those in English and the impact this may have had on the review

Please comment on your choice not to search any sources of grey literature, such as conference abstracts or dissertations, as recommended by the Cochrane Collaboration, https://handbook-5-1.cochrane.org/ Perhaps address this as you further address PRISMA checklist item 23c

PRISMA flow chart - in the first box of records excluded with reasons, there are 54 items excluded due to “Publication Type.” I recommend noting which publication types were excluded.

6. PLOS authors have the option to publish the peer review history of their article (what does this mean?). If published, this will include your full peer review and any attached files.

Reviewer #1: **Yes: **Erica R Brody

---

## [Author Response · Author response to Decision Letter 0]

16 Apr 2022

Editor 31st March, 2022

PLOS ONE

RE: PONE-D-21-18799

The Effects of Mindfulness-Based Interventions on Symptoms of Depression, Anxiety, and Cancer-Related Fatigue in Oncology Patients: A Systematic Review and Meta-Analysis

PLOS ONE

Dear Professor Sompop Bencharit, 

Below are our responses to each point made by the reviewers. As suggested, we have included the following documents in our revision:

• A ‘Response to Reviewers’ letter that responds to each point raised by the academic editor and reviewer(s);

• A marked-up copy of the manuscript titled ‘Revised Manuscript with Track Changes’

that highlights changes made to the original version; and an 

• unmarked version of our revised paper without tracked changes labelled 'Manuscript'.

Yours sincerely,

Litza Kiropoulos

Corresponding author (on behalf of all authors):

Litza Kiropoulos, PhD*

The University of Melbourne

Mood and Anxiety Disorders Lab

Melbourne School of Psychological Sciences

The University of Melbourne

Victoria 3010, Australia

Phone: (61) 03 9035 4063

E-mail: litzak@unimelb.edu.au

Response to reviewers:

1. In the Abstract: 

a) Search strategy, key words and time-line should be summarized in the Methods of the Abstract.

The search strategy, key words and timeline have now been stated in the Methods of the Abstract.

b) Please define your conclusion whether mindfulness reduces the symptoms from cancer or not.

We have now stated that the review found that MBIs reduced symptoms of depression, anxiety and CRF in oncology populations. 

2. In the Introduction:

Please clearly define these terms: Mindfulness-Based Stress Reduction18 (MBSR), Mindfulness-Based Cognitive Therapy (MBCT) and Mindfulness-Based Cancer Recovery (MBCR). 

Definitions have been given for MBSR, MBCT and MBCR (see lines 72-84).

3. in the Methods:

3.1 Define more clearly exclusion criteria. For example, I would imagine you exclude case reports, case series, reviews, etc. (and non-English literature?)

A statement about exclusion criteria has now been provided in the Methods section.

3.2 Did you do extensive search of the references for the reviews of full text articles? 

We have now placed a statement in the manuscript stating that we have undertaken an extensive search of the references for the review of full text articles (see page 129-130).

4.Please ensure that your manuscript meets PLOS ONE's style requirements, including those for file naming. 

The abstract and manuscript has been edited to meet the journal’s style requirements.

5. PLOS requires an ORCID iD for the corresponding author in Editorial Manager on papers submitted after December 6th, 2016. Please ensure that you have an ORCID iD and that it is validated in Editorial Manager. 

ORCID ID of corresponding author has now been validated in Editorial Manager for PLOS ONE and provided in this document above (ORCID ID https://orcid.org/ 0000-0002-1921-5904)

6. Please include a separate caption for each figure in your manuscript.

We have now included a separate caption for each figure in the manuscript. 

7. Please include your tables as part of your main manuscript and remove the individual files. Please note that supplementary tables (should remain/ be uploaded) as separate "supporting information" files.

We have now included all tables as part of the main manuscript and removed individual files. All supplementary files have been uploaded as separate ‘supporting information’ files. 

We have now included captions for the Supporting Information files at the end of the manuscript. 

9. Recommend more fully addressing item 7 of the PRISMA checklist to provide the full search strategy used for at least one database, including the keywords and controlled vocabulary with the Boolean operators used.

The full search strategy that has been used for all databases is presented in text on line 121-128.

10. Regarding PRISMA checklist item 23c - discuss any limitations of the review processes used - Please address this item more fully. For example, note that search results were limited to those in English and the impact this may have had on the review.

We have now added a statement about the limitations of the review process used in the manuscript. Specifically, we stated that the search results were limited to those written in the English language (see line 523-524).

11. Please comment on your choice not to search any sources of grey literature, such as conference abstracts or dissertations, as recommended by the Cochrane Collaboration, https://handbook-5-1.cochrane.org/ Perhaps address this as you further address PRISMA checklist item 23c.

We have now undertaken a search of the grey literature and searched all theses/dissertations using the same search terms. This resulted in one eligible dissertation being identified and this has now been included in the analyses.

12. PRISMA flow chart - in the first box of records excluded with reasons, there are 54 items excluded due to “Publication Type.” I recommend noting which publication types were excluded.

An explanation of ‘Publication Type’ has now been included on the PRISMA flow chart.

---

## [Editor Report · Decision Letter 1]

24 May 2022

The Effects of Mindfulness-Based Interventions on Symptoms of Depression, Anxiety, and Cancer-Related Fatigue in Oncology Patients: A Systematic Review and Meta-Analysis

PONE-D-21-18799R1

Dear Dr. Kiropoulos,

We’re pleased to inform you that your manuscript has been judged scientifically suitable for publication and will be formally accepted for publication once it meets all outstanding technical requirements.

Kind regards,

Sompop Bencharit, DDS, MS, PhD, FACP

Academic Editor

PLOS ONE

Additional Editor Comments (optional):

Thank you for the revision.
---

## [Editor Report · Acceptance letter]

16 Jun 2022

PONE-D-21-18799R1 

The Effects of Mindfulness-Based Interventions on Symptoms of Depression, Anxiety, and Cancer-Related Fatigue in Oncology Patients: A Systematic Review and Meta-Analysis 

Dear Dr. Kiropoulos:

I'm pleased to inform you that your manuscript has been deemed suitable for publication in PLOS ONE. Congratulations! Your manuscript is now with our production department. 

Kind regards, 

on behalf of

Dr. Sompop Bencharit 

Academic Editor

PLOS ONE